# Palmitic Acid Inhibits Myogenic Activity and Expression of Myosin Heavy Chain MHC IIb in Muscle Cells through Phosphorylation-Dependent MyoD Inactivation

**DOI:** 10.3390/ijms24065847

**Published:** 2023-03-19

**Authors:** Izumi Matsuba, Rikako Fujita, Kaoruko Iida

**Affiliations:** 1Department of Food and Nutritional Science, Graduate School of Humanities and Sciences, Ochanomizu University, 2-1-1 Otsuka, Bunkyo, Tokyo 112-8610, Japan; 2The Institute for Human Life Sciences, Ochanomizu University, 2-1-1 Otsuka, Bunkyo, Tokyo 112-8610, Japan

**Keywords:** palmitic acid, skeletal muscle, myosin heavy chain, MyoD, protein kinase C

## Abstract

Sarcopenia associated with aging and obesity is characterized by the atrophy of fast-twitch muscle fibers and an increase in intramuscular fat deposits. However, the mechanism of fast-twitch fiber-specific atrophy remains unclear. In this study, we aimed to assess the effect of palmitic acid (PA), the most common fatty acid component of human fat, on muscle fiber type, focusing on the expression of fiber-type-specific myosin heavy chain (MHC). Myotubes differentiated from C2C12 myoblasts were treated with PA. The PA treatment inhibited myotube formation and hypertrophy while reducing the gene expression of MHC IIb and IIx, specific isoforms of fast-twitch fibers. Consistent with this, a significant suppression of MHC IIb protein expression in PA-treated cells was observed. A reporter assay using plasmids containing the MHC IIb gene promoter revealed that the PA-induced reduction in MHC IIb gene expression was caused by the suppression of MyoD transcriptional activity through its phosphorylation. Treatment with a specific protein kinase C (PKC) inhibitor recovered the reduction in MHC IIb gene expression levels in PA-treated cells, suggesting the involvement of the PA-induced activation of PKC. Thus, PA selectively suppresses the mRNA and protein expression of fast-twitch MHC by modulating MyoD activity. This finding provides a potential pathogenic mechanism for age-related sarcopenia.

## 1. Introduction

Sarcopenia is an age-related, progressive, and generalized skeletal muscle disorder characterized by low muscle strength and mass and reduced functional performance [1,2]. It becomes a major cause of frailty, which dramatically affects health status and quality of life in the elderly [3,4]. Sarcopenia is caused by a variety of factors, including not only aging but also poor nutrition, reduced physical activity, and chronic diseases [1,2]. Obesity, especially age-associated obesity, is closely involved in the development of sarcopenia [5]. A reduced muscle mass correlates with a lower resting metabolic rate, which perpetuates the development of obesity. Indeed, in adult males, muscle mass contributes to 30–42% of the total body weight in younger individuals, declining with age to 18–27%, while the body fat mass increases from 20% to 30% [6,7]. Thus, lipid deposition occurs in various organs, including skeletal muscles, leading to the metabolic disturbance known as lipotoxicity [8,9]. In the elderly, lipotoxicity acts synergistically with sarcopenia to cause various chronic diseases, such as cardiovascular diseases [10,11].

Saturated fatty acids, including palmitic acid (PA), are thought to play a causal role in lipotoxicity [12]. PA, the most common and predominant saturated fatty acid in the human body [13], is generally stored as a component of triglycerides in lipid droplets in adipose tissue. However, when lipid accumulation in adipose tissue or lipid intake from diet is excessive, the PA in the blood also increases, acting in a paracrine or endocrine manner in various tissues, with adverse effects that lead to disease development [14,15]. In many in vitro and in vivo experimental settings, PA treatment has been shown to induce inflammatory reactions, mitochondrial dysfunction, and the generation of reactive oxygen species, leading to cellular damage and cell and tissue death [13,15,16]. Several in vitro studies evaluating the effects of PA on myocytes have reported that PA exposure negatively affects myocyte maintenance by suppressing protein synthesis [17] and upregulating proteolytic signaling [18]. Furthermore, PA has been shown to induce endoplasmic reticulum (ER) stress, which in turn triggers programmed cell death via apoptosis [19,20]. PA also regulates the expression of several microRNAs, thereby inhibiting myoblast differentiation and maturation into myofibers [21,22].

Skeletal muscles are comprised of two types of fibers with distinct metabolic profiles: glycolytic (type II) fibers that mainly use glucose as a substrate, and the more oxidative (type I) fibers that use lipids as a substrate. Reductions in muscle mass associated with aging are known to occur mainly in type II fibers [23,24]. This raises the question of whether intramuscular lipid accumulation, particularly excess PA, plays a causal role in the selective reduction of type II fibers in the elderly. Carter et al. [25] hypothesized that type II fibers are more vulnerable to lipotoxic stress based on findings that type II fibers have less cellular machinery to deal with the fatty acid spillover than type I fibers [26]. This is supported by animal studies that demonstrated a greater adaptive response of type I fibers to intracellular lipid accumulation [27].

Muscle fiber types are determined by the expression of different myosin heavy chain (MHC) isoforms, MHC I, IIa, IIx, and IIb, with type I fibers primarily expressing MHC I and type II fibers primarily expressing MHC IIx and IIb (although MHC IIb is rarely detected in human muscle) [28]. MHC gene expression is dynamic and changes with specific conditions such as training, immobility, or disease. A significant reduction in the mRNA levels of MHC IIx, a predominant isoform in human type II fibers, with no change in MHC I, has been reported in aging [29,30]. We therefore further hypothesized that increased PA due to intramuscular lipid accumulation is directly responsible for reduced MHC II mRNA expression.

Therefore, in the present study, we investigated whether PA differentially affects the MHC mRNA expression of each MHC isoform, using muscle cells derived from the C2C12 mouse myoblast cell line.

## 2. Results

### 2.1. PA Inhibits the Terminal Differentiation of Myoblasts into Myotubes 

We initially examined the effect of PA on morphological changes and MHC production in differentiated C2C12 cells. In cells treated with 400 μM PA on day 5 for 24 h (Figure 1a), the ratio of differentiated myotubes expressing MHC, a marker of terminal differentiation, was lower than in the untreated control cells (Figure 1b). The calculated fusion index and the average amount of MHC in the cultures treated with PA were significantly lower than in the untreated control at the same time point (Figure 1c).

### 2.2. PA Primarily Inhibits the Expression of MHC IIb in Differentiated Cells

The effect of PA on the expression of each MHC gene was examined. Mouse muscles contain four MHC isoforms (MHC I, IIa, IId/x, and IIb) [28], and the transcripts of all these isoforms are well expressed in C2C12 cells after day 4 of differentiation [31]. Hence, the effect of PA on the expression of genes encoding each MHC isoform was examined on days 4 and 6 (Figure 2a). Treatment with 400 μM PA for 24 h significantly suppressed the expression of *Myh4* (encoding MHC-IIb) on days 4 and 6 (Figure 2b) and that of *Myh1* (encoding MHC-IId/x) on day 6 (Figure 2b). In contrast, the expression levels of *Myh2* (encoding MHC-IIa) and *Myh7* (encoding MHC-I) were not altered by PA treatment on either day. These effects of PA on *Myh4* expression were observed from the early to late stages of differentiation (from day 1 to day 6) (Figure 2c). As the expression of MHC genes is strictly controlled by several myogenic regulatory factors (MRFs), the levels of *Myod and Myog* (encoding MyoD and myogenin, respectively) were determined on days 4 and 6. The results showed that *Myod* expression was significantly suppressed by PA treatment on both days, whereas *Myog* expression was suppressed only on day 6 (Figure 2d).

The inhibition of MHC protein expression by PA was further confirmed by Western blotting. Consistent with the gene expression results, C2C12 myotubes treated with 400 μM PA on day 6 contained considerably less MHC IIb protein than untreated controls (Figure 3). The MHC I protein expression was also reduced by PA, inconsistent with the results of the gene expression analysis (Figure 3). 

### 2.3. The Inhibitory Effect of PA on MHC Expression in Myotubes Is Not Mediated via TLR-4

PA is known to act as a ligand for toll-like receptors (TLR), especially TLR4, which promotes inflammatory responses [32]. To investigate whether the activation of this receptor is involved in the PA-induced inhibition of myotube differentiation and *Myh4* expression, we assessed the effect of a TLR4-specific inhibitor, TAK-244. TAK-244 had no influence on the inhibitory effects of PA on cellular MHC production, based on fluorescence intensity (Figure 4a) and *Myh1* and *Myh4* expression in C2C12 myotubes (Figure 4b). 

### 2.4. PA Suppresses MyoD-Induced Activation of Myh4 Promoter in C2C12 Cells

Using a luciferase reporter assay, we further examined whether PA reduced MHC gene expression by directly suppressing promoter activity, focusing specifically on the *Myh4* gene. The transfection of C2C12 myoblasts with the luciferase reporter gene under the control of the mouse *Myh4* promoter resulted in increased reporter gene transcription during cell differentiation (Figure 5a). Treatment with 400 μM PA for 24 h significantly reduced reporter gene transcription relative to the vehicle control on each day of differentiation (Figure 5a).

The *Myh4* promoter is preferentially activated by MyoD [33]. Therefore, we examined whether MyoD transcriptional activity was suppressed by PA treatment. Promoter activity of the pGL-MHCIIb reporter construct was substantially elevated in undifferentiated myoblasts under the forced expression of MyoD with pcDNA–MyoD relative to that in cells transfected with the empty pcDNA3 vector (Figure 5b). After PA treatment for 24 h, promoter activity was also upregulated through the forced expression of MyoD, albeit moderately compared to the activity in the untreated cells (Figure 5b).

### 2.5. The Inhibitory Effect of PA on Myh4 Promoter Activation by MyoD Requires Ser Residues That Are Targets for Phosphorylation

The MyoD protein contains several phosphorylation sites, and the phosphorylation of specific sites, such as Thr^115^ [34] or Ser^200^ [35,36], regulates its myogenic function. Therefore, we examined whether the phosphorylation of MyoD is involved in the mechanism by which PA suppresses the MyoD-induced increase in *Myh4* promoter activity.

The mutant forms of MyoD expressed by each expression plasmid are shown in Figure 6a. In undifferentiated myoblasts expressing MyoD-T/A with a single mutation at Thr^115^, the PA-induced inhibition of *Myh4* promoter activity was statistically recovered. However, the promoter activity was significantly suppressed in cells expressing this mutant compared with native MyoD-expressing controls (Figure 6b). Conversely, in myoblasts expressing MyoD-S/A with a single mutation on Ser^200^, promoter activity was significantly suppressed by PA treatment to a similar extent as in control cells expressing native MyoD (Figure 6c). When the mutant MyoD-S/A-2, with mutations at Ser^5^ and Ser^262^ in addition to Ser^200^ (all known phosphorylation targets [37,38]), was expressed in cells, *Myh4* promoter activity was elevated relative to the activity in the control cells (Figure 6d). The PA-induced reduction in *Myh4* promoter activity was still observed in MyoD-S/A-2-expressing cells; however, this decrease was significantly recovered compared to the control cells expressing native MyoD (Figure 6d).

### 2.6. Inhibition of PKC Partially Recovers PA-Induced Inhibition of MHC IIb Gene Expression in C2C12 Myotubes

The phosphorylation of MyoD is likely to be involved in the inhibition of *Myh4* transcription by PA. Therefore, we tested whether kinase inhibition attenuated the suppressive effect of PA on the expression of *Myh4*. For validation, we used the following three inhibitors: (a) Ro 31-8220, a potent PKC inhibitor of PKCα/β/γ and PKCε; (b) myriocin, an SPT inhibitor that inhibits the conversion of PA to ceramide, an activator of PKCζ; (c) roscovitine, a selective CDK inhibitor of CDK1 and CDK2. We observed that 0.5 μM Ro 31-8220 partially diminished the suppression of *Myh4* expression by PA treatment (Figure 7a). In contrast, myriocin (Figure 7b) and roscovitine (Figure 7c) at up to 5 μM did not affect the PA-induced suppression of *Myh4* expression. 

Finally, Figure 8 summarizes the putative mechanism by which PA selectively suppresses the expression of fast-twitch MHC, MHC IIb. 

## 3. Discussion

In this study, we found that PA suppressed the mRNA and protein expression of MHC, specifically MHC IIb, a predominant form of fast-twitch fiber, in C2C12 myotubes by inhibiting the transcriptional activity of MyoD. We suggest that the mechanism by which PA inhibits MyoD transcriptional activity is the phosphorylation of MyoD, and that a specific type of PKC participates in this mechanism. Several studies have examined the direct effects of PA on cultured myoblasts and myocytes. Similar to the findings of the present study, treatment of myoblasts with PA has been shown to inhibit myogenic cell differentiation and myotube maturation [21,22,39,40] and reduce type II myosin protein expression [41]. We further clarified the detailed mechanism by which PA treatment selectively inhibits the expression of specific MHC isoforms at a transcription level during myoblast differentiation. 

TLR4 is a major receptor that recognizes pathogen-associated molecular patterns, leading to the induction of sterile inflammation. PA cross-reacts with this receptor and causes the derangement of intracellular signaling pathways in muscle cells and tissues [32,42,43]. Several previous studies have suggested that the pro-inflammatory effects of PA and the associated inhibition of PI3K/Akt signaling are involved in the mechanism by which PA reduces the differentiation potential of myogenic cells [39,40]. However, in the present study, the TLR4-specific inhibitor TAK-242 did not affect the PA-induced inhibition of *Myh4* expression, suggesting that the signaling pathway involving TLR4 is independent of the mechanism.

The process of myogenic differentiation is strictly regulated by several transcription factors, including Myf5, MyoD, and myogenin. Of these, MyoD and Myf5 are required for commitment to the myogenic lineage in the early stage of differentiation. In contrast, myogenin is believed to play a critical role in the expression of the terminal muscle phenotype pre-established by MyoD late in the differentiation process [44]. MyoD transcriptionally activates myogenin expression and autoactivates its own expression through a positive feedback loop. Moreover, several studies have confirmed that MyoD plays a crucial role in MHC IIb expression in mature muscles [33,45,46]. In the denervated soleus muscles of rats and mice, the overexpression of normal MyoD, or a mutant form resistant to inactivation, increased the number of fast-type MHC fibers [47]. In the present study, PA treatment specifically suppressed the expression of MHC IIb, a fast-type MHC isoform. This suppressive effect was observed during the early stage of myoblast differentiation (on day 1). In addition, *Myod* and *Myog* gene expression was suppressed by PA treatment. Based on these observations, we focused on MyoD as a key molecule in the mechanism by which PA regulates gene expression in MRFs and MHCs. As expected, the results of the reporter assay suggest that *Myh4* expression is repressed at the transcriptional level by PA during differentiation and that, as a mechanism, PA administration suppresses the MyoD-dependent transcription of *Myh4*. 

The MyoD protein contains several phosphorylation sites targeted by specific protein kinases, such as PKC and CDK, and its activity is thought to be regulated by post-transcriptional modification. For instance, phosphorylation of serine residues at positions 5, 200, and 262 by CDK1 and/or CDK2 results in the degradation of the MyoD protein [35,36,37,38]. Another study reported that two putative PKC phosphorylation sites, Ser^200^ and Thr^115^, are required for MyoD inactivation [34]. Here, MyoD, with amino acid substitutions of serine residues that disrupt phosphorylation by CDK and PKC, significantly recovered the PA-induced inhibition of *Myh4* promoter activity. The amino acid substitution at Thr^115^ also recovered PA-induced reduction in *Myh4* promoter activity. However, this mutant severely suppressed *Myh4* promoter activity itself, which is inconsistent with previous results that demonstrated that the expression of mutant (Thr^115^ to Ala) MyoD leads to an increase in MHC protein expression in mouse fibroblasts [34]. The cause of this discrepancy between studies is unknown; however, a reduction in promoter activity by this mutant may be due to the importance of Thr^115^ in the binding capacity of MyoD [48].

Finally, the experimental results using a specific inhibitor showed that PKC, but not CDK, is likely to be involved in the mechanism of the PA-induced inhibition of *Myh4* transcription. PKCs form a large family of serine–threonine kinases that can be classified into three subfamilies, depending on the specific lipids required for their activation. Conventional PKCs (cPKCs, including PKCα/β/γ) and novel PKCs (nPKCs, including PKCε) require a lipid second messenger, diacylglycerol (DAG), for activation; the former require Ca^2+^ and the latter are Ca^2+^-independent, whereas atypical PKCs (aPKCs, such as PKCζ) are not activated by DAG but by another lipid mediator, sphingolipid ceramide [49,50,51]. Both DAG and ceramide are synthesized from PA; however, Ro 31-8220, not myriocin, restored PA-induced reductions in *Myh4* expression, suggesting the involvement of DAG-activated PKC in the mechanism. Levels of DAG were significantly higher in myogenic cells treated with PA and in the muscles of genetically obese mice than in controls [52,53]. Such an elevation in DAG levels would lead to the suppression of MyoD activity through phosphorylation by PKC.

On the other hand, we could not determine whether Ro 31-8220 could completely eliminate the PA-induced inhibition of *Myh4* expression because MCH gene expression was severely suppressed when this inhibitor was used at high concentrations (Appendix A). Ro 31-8220 inhibits the activity of PKCε, which has been reported to play important roles in myoblast differentiation [54,55]. Therefore, treatment with this inhibitor may lead to the suppression of MHC expression. In addition, a previous study suggested the involvement of PKCθ, a member of nPKCs, in the biological effects of PA on C2C12 myocytes [56]. Further studies are needed to ascertain which PKCs are more critical for the action mechanism of PA.

In recent years, sarcopenia has become a major health problem in an aging society. The histological changes in age-related loss of muscle tissue are characterized by the atrophy of fast-twitch muscle fibers and an increase in intramuscular fat deposition [57]. The present study reveals that PA, a predominant component of body fat, selectively suppresses the mRNA and protein expression of fast-twitch MHC by modulating the transcriptional activity of MyoD, suggesting a pathogenic mechanism of sarcopenia.

## 4. Materials and Methods

### 4.1. Materials 

PA was purchased from Nacalai Tesque (Kyoto, Japan). Fatty-acid-free bovine serum albumin (BSA) was purchased from FUJIFILM Wako Chemicals (Osaka, Japan). TAK-242, an inhibitor of toll-like receptor 4 (TLR4), and myriocin, an inhibitor of serine palmitoyl-transferase (SPT), a key enzyme for ceramide de novo synthesis, were purchased from Cayman Chemical (Ann Arbor, MI, USA). Ro 31-8220, a potent inhibitor of protein kinase C (PKC), was obtained from Santa Cruz Biotechnology (Dallas, TX, USA). Roscovitine, a selective cyclin-dependent kinase (CDK) inhibitor of CDK1 and CDK2, was purchased from FUJIFILM Wako Chemicals. The inhibitors were reconstituted in dimethyl sulfoxide. 

### 4.2. Palmitic Acid Complex Preparation

PA was complexed with BSA in phosphate-buffered saline (PBS). PA was first added to a 100 mM NaOH solution at a concentration of 100 mM and dissolved on a heat block at 75 °C for 30 min. The prepared PA solution was then added to 10% (*w*/*v*) fatty-acid-free BSA in PBS at a ratio of 1:9 (*v*/*v*) to obtain a PA–BSA solution with a PA concentration of 10 mM. This solution was added to the medium at a final concentration of 400 μM PA. The chosen concentration of PA is within the physiological range of the plasma PA concentration of healthy adults [58]. For the vehicle control, a mixture of 100 mM NaOH and 10% BSA/PBS (1:9) was used. Each prepared solution was warmed to 55 °C before being added to the cells.

### 4.3. Cell Culture and Treatment

C2C12 myoblasts (RIKEN, Tsukuba, Japan) were cultured in Dulbecco’s modified Eagle’s medium (DMEM) containing 10% fetal bovine serum at 37 °C in a humidified atmosphere with 5% CO_2_. Myoblasts were seeded on the type-I-collagen-coated plate (Iwaki, Shizuoka, Japan) at a density of 3.0 × 10^4^ cells/cm^2^. The following day (day 0), differentiation into myotubes was initiated by replacing the medium with a differentiation medium (DMEM containing 2% horse serum). The medium was changed daily until the cells were collected for analysis. In the treated set, the C2C12 myotubes were incubated with 400 μM PA from the indicated day of differentiation (day 0 to day 5) for 24 h. In some of the experiments, the inhibitors (1 μM TAK-242, 0.5 μM Ro 31-8220, 5 μM myriocin, or 5 μM roscovitine) were added separately with the PA. We confirmed that inhibitor treatment did not affect cell viability at the indicated concentrations (data not shown). 

### 4.4. Gene Expression Analysis Using Real-Time PCR

The total RNA was isolated from cells using Sepasol-RNA I reagent (Nacalai Tesque). First-strand cDNA was synthesized using ReverTra Ace qPCR RT Master Mix (TOYOBO, Osaka, Japan), following the manufacturer’s instructions. Next, the quantitative real-time PCR was carried out using a SYBR premix Ex Taq II (TAKARA BIO, Shiga, Japan) with 10 µL reactions on a Thermal Cycler Dice Real Time System (TAKARA BIO) for amplification. The PCR cycling conditions were one cycle of 30 s at 95 °C, followed by 40 cycles of 5 s at 95 °C and 30 s at 60 °C. Gene mRNA expression was normalized to that of a standard housekeeping gene (*Gapdh*) using the ΔΔCT method. Experiments were performed in duplicate and repeated three times independently. The primer sequences are shown in Appendix A.

### 4.5. Western Blotting

The cells were washed with PBS, dissolved in lysis buffer (1% Triton X-100, 0.45% sodium pyrophosphate, 100 mM NaF, 2 mM Na_3_VO_4_, 50 mM HEPES (pH 8.0), 147 mM NaCl, 1 mM EDTA, and protease inhibitor mixture (cOmplete; Sigma-Aldrich, Tokyo, Japan), and centrifuged at 12,000× *g* for 15 min at 4 °C. The supernatant was used as the cell lysate. Referring to a previous report [59], cell lysate proteins (50 μg per lane) were resolved on 8% SDS-PAGE gel containing 30% glycerol, and electrophoresed at 100 V for 24 h at 4 °C (this electrophoretic condition allows for the separation of MHC isoforms on the same gel). The protein bands were then transferred to a PVDF membrane (Hybond-P; GE Healthcare Life Science, Tokyo, Japan). The membrane was blocked with Blocking One P (Nacalai Tesque) for 30 min at room temperature (20–25 °C) and incubated overnight at 4 °C with an antibody against MHC (1:1000; MF-20; Hybridoma Bank, Iowa, IA, USA), which recognizes all MHC isoforms, or with β-actin (1:1000; sc-47778; Santa Cruz Biotechnology). Next, the membranes were incubated with a secondary anti-mouse IgG antibody (1:5000; #7076; Cell Signaling Technology, Danvers, MA, USA) in Tris-buffered saline containing 0.1% Tween 20 for 1 h at room temperature. The bands were developed using an ECL Prime Western Blotting Detection Reagent (GE Healthcare Life Science), and images were captured using an ImageQuant LAS-4000 (GE Healthcare Life Science) image analyzer. The optical density of each band was analyzed using ImageJ software (National Institutes of Health, Bethesda, MD, USA). Experiments were repeated twice independently.

### 4.6. Immunofluorescence and Calculation of the Myogenic Index

The differentiated C2C12 cells were fixed and permeabilized using 1% Triton-X containing 4% paraformaldehyde in PBS for 10 min at room temperature. The cells were then incubated with an MF-20 anti-MHC antibody (1:100) for 90 min at room temperature, washed with PBS containing 0.1% Tween 20 (PBS-T), and incubated for 1 h with anti-mouse IgG conjugated with fluorescein isothiocyanate (FITC) (1:500; #115-095-062; Jackson ImmunoResearch, West Grove, PA, USA). After washing the samples with PBS-T, the nuclei were stained with 4,6-diamidino-2-phenylindole (DAPI). For each condition, 15 images of the cells (n = 3, five images per sample) were randomly taken using the 40× objective lens on a fluorescence microscope (BZ-X700; Keyence, Osaka, Japan). The number of nuclei in each sample was counted from the photographs, and the MHC-positive polynucleic cells stained with green fluorescence were defined as myotubes. The fusion index (%) was calculated using the following formula: (number of nuclei within myotubes/total number of nuclei) × 100. The average myocellular MHC content was estimated by dividing the total green fluorescence intensity (in pixels) by the total number of nuclei. The measurement methods are summarized in Appendix A.

### 4.7. Reporter Assays

The luciferase reporter plasmid, under the control of the *Myh4* promoter, and the MyoD expression plasmid were previously constructed [60]. Briefly, mouse promoter sequences (−1347 to +33) of *Myh4* (encoding MHC IIb protein) were amplified from mouse genomic DNA using specific primers with suitable restriction sites. These were cloned into the pGL3 basic vector (Promega, Tokyo, Japan) and named pGL-MHCIIb. Full-length mouse MyoD cDNA was amplified by PCR, using specific primers with suitable restriction sites, and cloned into a pcDNA3 mammalian expression vector named pcDNA-MyoD. Using the pcDNA-MyoD and KOD-Plus-Mutagenesis Kit (Toyobo, Osaka, Japan), three additional mutant MyoD expression plasmids were constructed: those expressing a MyoD protein that substituted alanine (Ala) at each of the following positions: threonine (Thr) residue Th^115^ (pcDNA-MyoD-T/A), serine (Ser) residue S^5^ (pcDNA-MyoD-S/A), and multiple serine residues S^5^, S^200^, and S^262^ (pcDNA-MyoD-S/A-2). One day before transfection, C2C12 cells (3 × 10^4^ cells/well) were seeded into 24-well plates (non-coated for undifferentiated cells, or collagen-coated for differentiated cells), and the indicated plasmid and control plasmid (pRL-SV40) (Promega), containing early SV40 enhancer/promoter region upstream of the Renilla luciferase gene, were co-transfected into cells using X-tremeGENE HP DNA Transfection Reagent (Sigma-Aldrich), in accordance with the manufacturer’s protocol. For differentiated samples, the growth medium was changed to the differentiation medium the next day. The transfected cells were then incubated with or without PA for 24 h until the cells were harvested, and the luciferase activity was measured using the Dual-Luciferase Reporter Assay System (Promega). The values were expressed as a fold induction and were corrected for transfection efficiency using Renilla luciferase activity. The experiments were performed in duplicate and were repeated three times independently.

### 4.8. Statistical Analysis

The data are expressed as the mean ± standard error of the mean (SEM). All statistical analyses were performed using SPSS Statistics for Windows (version 24; SPSS Inc., Chicago, IL, USA). An unpaired Student’s *t*-test was used to identify significant differences between the two groups. A one-way ANOVA, followed by Tukey’s post-hoc test, was performed to determine differences among three or more groups. A two-way ANOVA was used to determine interactions between two independent variables. Differences were considered statistically significant at *p* < 0.05.

## Figures and Tables

**Figure 1 ijms-24-05847-f001:**
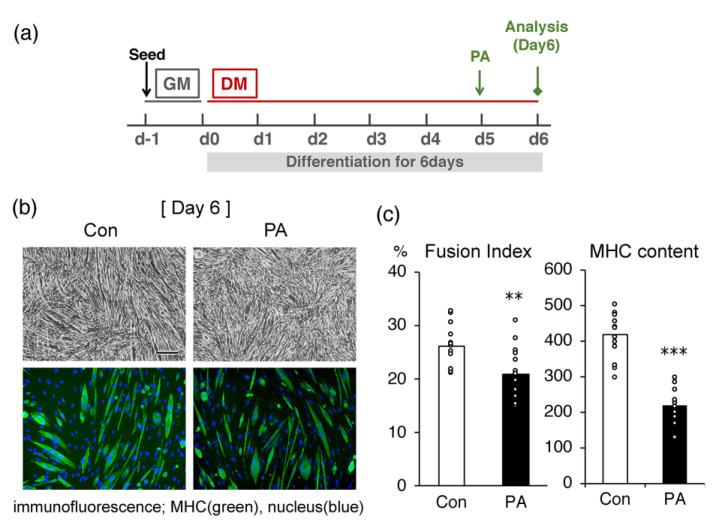
Palmitic acid (PA) inhibits the myogenic differentiation of C2C12 myoblasts. Cells were incubated with differentiation medium (DMEM containing 2% horse serum) for 5 days, then additionally incubated for 24 h in the absence (control; Con) or presence (PA) of 400 μM PA. (**a**) Time schedule for the experiment. The cells were seeded with growth medium (GM) one day beginning the experiment (d−1). The next day (d0), the medium was replaced with a differentiation medium (DM) to promote cell differentiation. The PA was added on day 5 (d5) for 24 h. (**b**) Representative microscopic images of the bright field (upper panel) or immunostaining (lower panel) of C2C12 cells. Scale bar, 200 μm. (**c**) The fusion index (%) (left), the number of nuclei located within MHC-positive myotubes divided by the total number of nuclei and expressed as a percentage, and average MHC content (pixels) (right), the green fluorescence intensity divided by the total number of nuclei. The bar graphs indicate the average (n = 15 images). Each circle in the graph represents an individual data point. ** *p* < 0.01; *** *p* < 0.001, versus the control.

**Figure 2 ijms-24-05847-f002:**
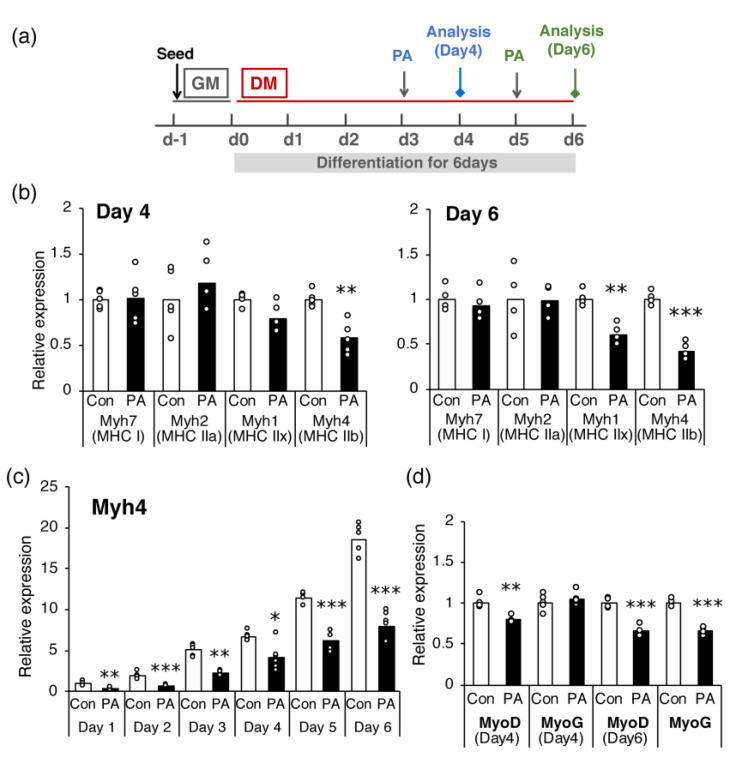
Palmitic acid (PA) inhibits the expression of specific MHC genes, *Myh1* and *Myh4*, and MyoD and myogenin, in differentiated cells. C2C12 cells were differentiated and incubated with a medium containing a vehicle control (Con) or 400 μM PA (PA) for 24 h prior to cell harvest. Cells were harvested on the indicated day of differentiation, and mRNA expression levels of MHC, MyoD, and myogenin (MyoG) were determined using real-time PCR. (**a**) Time schedule for the experiment. The cells were seeded with growth medium (GM) one day before starting the experiment (d−1). The next day (d0), the medium was replaced with a differentiation medium (DM) to promote cell differentiation. PA was added 1 day before cell harvest. The expression levels of (**b**) each MHC gene on days 4 and 6, (**c**) MHC IIb gene (*Myh4*) on days 1–6, and (**d**) MyoD and MyoG on days 4 and 6 were quantified and normalized to that of *Gapdh*. Values are expressed as fold-change relative to the control (**b**,**d**), or to the control on day 1 (**c**), with each value used for normalization arbitrarily set to 1. The bar graphs indicate the average (n = 4–6 per group). Each circle in the graph represents an individual data point. * *p* < 0.05; ** *p* < 0.01; *** *p* < 0.001, versus each control.

**Figure 3 ijms-24-05847-f003:**
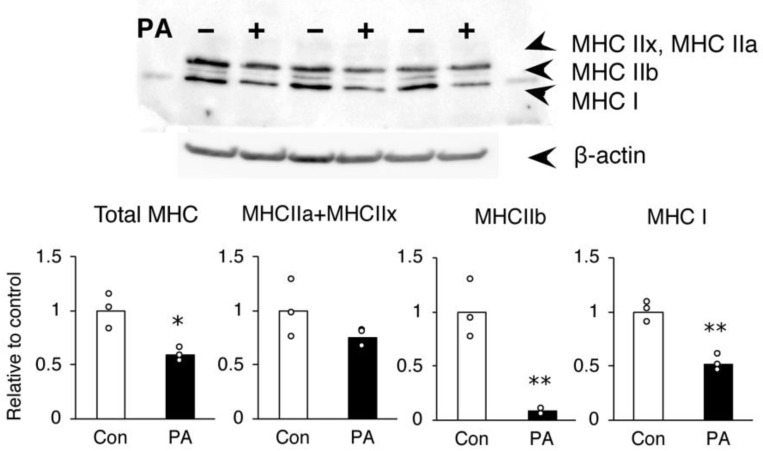
Palmitic acid (PA) inhibits the expression of specific MHC proteins in C2C12 myotubes. C2C12 cells were differentiated and incubated for the last 24 h of differentiation with a medium containing a vehicle control (Con) or 400 μM PA (PA). The cells were harvested on day 6, and protein levels of MHC and β-actin were determined using immunoblotting. Levels were quantified and expressed as fold-change relative to the control. The bar graphs indicate the average (n = 3 per group). Each circle in the graph represents an individual data point. * *p* < 0.05; ** *p* < 0.01, versus each control.

**Figure 4 ijms-24-05847-f004:**
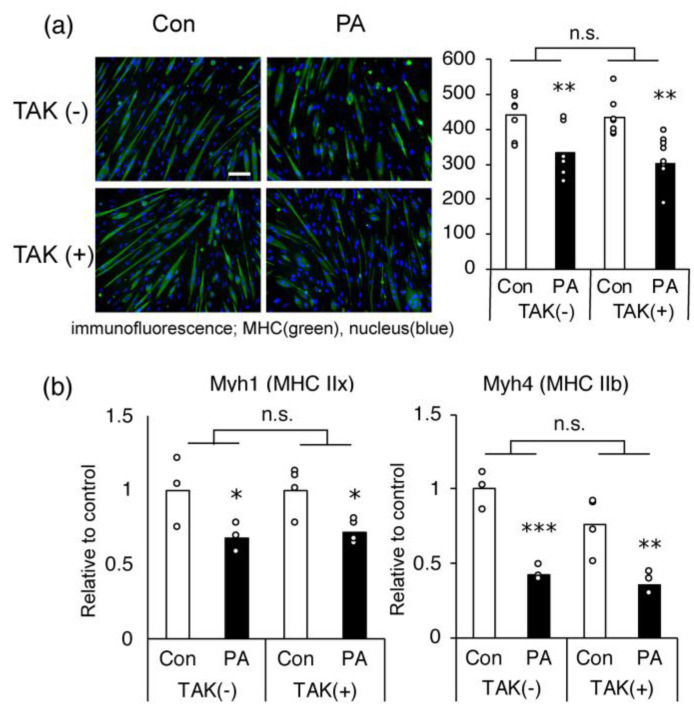
The Toll-like receptor (TLR)-related pathway is not involved in palmitic acid (PA)-induced inhibition of myogenesis and MHC gene expression in C2C12 cells. Differentiated cells on day 5 were further incubated for 24 h in the absence (control, Con) or presence (PA) of 400 μM PA, with or without 1 μM TAK-242 (TAK), a TLR4-specific inhibitor. (**a**) Representative microscopic images of immunostaining and average MHC content (in pixels; green fluorescence intensity divided by the total number of nuclei, n = 8 per group). Scale bar, 200 μm. (**b**) The mRNA expression levels of *Myh1* and *Myh4* on day 6 were quantified and normalized to that of *Gapdh*. Values are expressed as fold-change relative to the control (n = 4 per group). The bar graphs indicate the average. Each circle in the graph represents an individual data point. * *p* < 0.05; ** *p* < 0.01; *** *p* < 0.001, versus each control group. n.s.: no significant interaction by two-way ANOVA.

**Figure 5 ijms-24-05847-f005:**
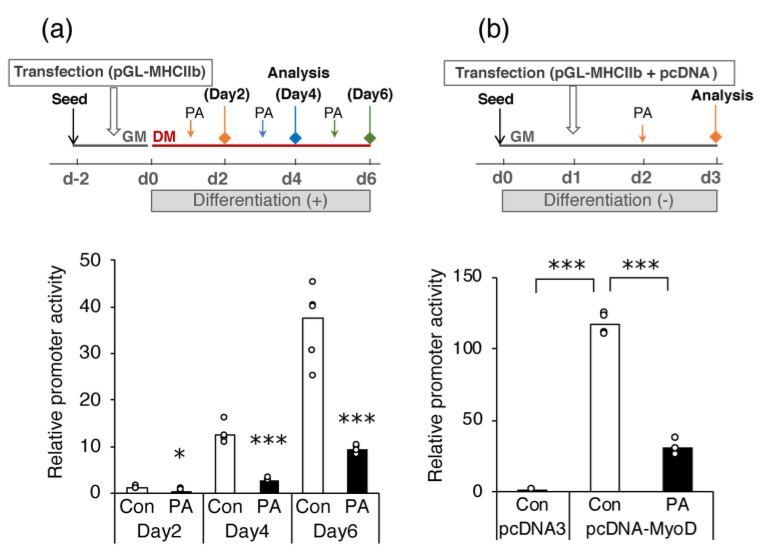
Palmitic acid (PA) reduces the activation of *Myh4* transcription via a mechanism involving MyoD. (**a**) Myoblasts were seeded with growth medium (GM) two days before starting the experiment (d−2). The cells were transfected with the pGL-MHCIIb plasmid and control plasmid (pRL-SV40) one day before the initiation of differentiation (d−1). The transfected cells were differentiated for the indicated number of days in the presence of the vehicle control (Con) or 400 μM PA (PA) for the last 24 h. (**b**) Myoblasts were seeded with growth medium (GM); the next day, the cells were transfected with the pGL-MHCIIb, pRL-SV40, and each of the following expression plasmids, pcDNA3 and pcDNA-MyoD. After overnight incubation, the cells were treated with the vehicle control or 400 μM PA (PA) for an additional 24 h. Luciferase activity was measured and normalized to Renilla luciferase activity to determine transfection efficiency. Values are expressed as the fold-change relative to the vehicle control cells on day 2 (d2) (**a**), or to the cells transfected with pcDNA3 (**b**), with each value used for normalization arbitrarily set to 1. The bar graphs indicate the average (n = 4–6 per group). Each circle in the graph represents an individual data point. * *p* < 0.05; *** *p* < 0.001, versus each control group in (**a**).

**Figure 6 ijms-24-05847-f006:**
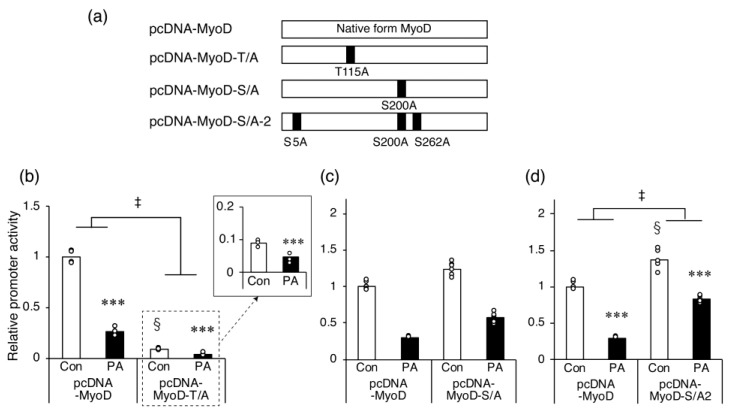
Disruption of MyoD phosphorylation diminishes palmitic acid (PA)-induced inhibition of *Myh4* promoter activity. (**a**) Schema of the wild-type and mutant forms of MyoD expressed by each expression plasmid. (**b**–**d**) Myoblasts were seeded one day before transfection, then co-transfected with pGL-MHCIIb, pRL-SV40, and (**b**) pcDNA-MyoD-T/A, (**c**) pcDNA-MyoD-S/A, or (**d**) pcDNA-MyoD-S/A-2. After overnight incubation, the vehicle control (Con) or 400 μM PA (PA) was added to the medium for 24 h. Luciferase activity was measured and corrected to that of Renilla luciferase activity. Values are expressed as fold-change relative to the cells in the vehicle control transfected with pcDNA-MyoD. Each value used for normalization was arbitrarily set to 1. The bar graphs indicate the average (n = 5–6 per group). *** *p* < 0.001, versus each control. § *p* < 0.001, versus the vehicle control transfected with pcDNA-MyoD. Each circle in the graph represents an individual data point. ‡ significant interactions between groups (*p* < 0.05 by two-way ANOVA).

**Figure 7 ijms-24-05847-f007:**
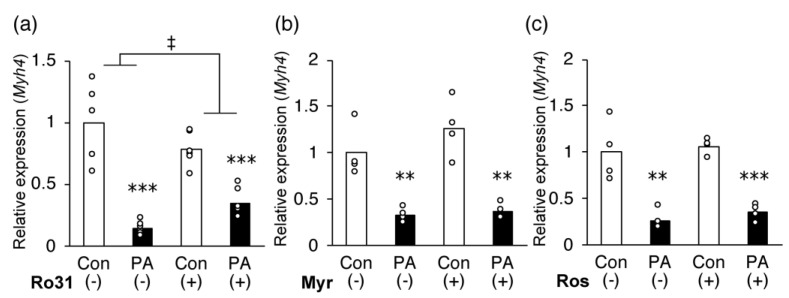
A protein kinase C (PKC)-specific inhibitor restores the palmitic acid (PA)-induced reduction in *Myh4* expression. C2C12 cells were differentiated for 5 days and incubated for an additional 24 h with the medium containing the vehicle control (Con) or 400 μM PA (PA) with or without the specific inhibitor (**a**) 0.5 μM Ro 31-8220 (Ro31), (**b**) 5 μM myriocin (Myr), or (**c**) 5 μM roscovitine (Ros). The mRNA expression of *Myh4* on day 6 was quantified and normalized to that of *Gapdh*. Values are expressed as fold-change relative to the control without inhibitor. The results are expressed as mean ± SEM (n = 4 per group). *** *p* < 0.001; ** *p* < 0.01, versus each control. ‡ significant interaction between groups (*p* < 0.05 by tow-way ANOVA).

**Figure 8 ijms-24-05847-f008:**
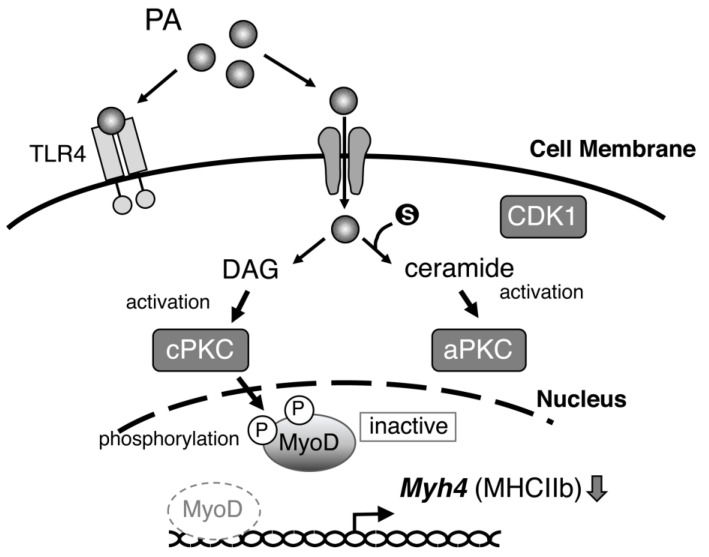
Putative mechanisms of the inhibitory effect of palmitic acid (PA) on MHC IIb expression in muscle cells. CDK1: cyclin-dependent kinase 1; DAG: diacylglycerol; aPKC: atypical PKC; cPKC: conventional PKC; TLR4: toll-like receptor 4, a receptor for the recognition of pathogen-associated molecular patterns. DAG derived from PA might activate cPKC and thereby de-phosphorylate MyoD, leading to its inactivation. aPKC activated by PA-derived ceramide is likely not involved in the mechanism.

## Data Availability

The data that support the findings of this study are available from the corresponding author upon reasonable request.

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
