# Peer review of "Palmitic Acid Inhibits Myogenic Activity and Expression of Myosin Heavy Chain MHC IIb in Muscle Cells through Phosphorylation-Dependent MyoD Inactivation"

_ijms, 2023, doi:10.3390/ijms24065847_

Round 1
Reviewer 1 Report
This manuscript from Matsuba et al. examines the effect of palmitic acid (PA) on myosin heavy chain IIb (and other isoforms) along with some myogenic regulators, including MyoD. The manuscript is clearly written, with a straightforward experimental design. The authors conclude that PA inhibits mRNA and protein expression for MHCIIb, putatively through decreased PKC-MyoD signaling. Although the manuscript is interesting, I have several comments and concerns regarding its suitability for publication, as listed below.
1) The novelty of the data in this paper is lacking. Nguyen et al. (IJMS 21:9445, 2020, Reference 21) previously demonstrated that PA inhibits differentiation, decreases myosin heavy chain protein expression in general, and suppresses MyoD and MyoG expression in C2C12 myotubes. PA specifically reduces protein expression of type II myosin (Fan et al., Life Sciences, https://doi-org.byu.idm.oclc.org/10.1016/j.lfs.2020.118243; the antibody used detects type II MHC). PKC activity is already known to be involved in PA action in myotubes (Jove et al., 2006; https://doi.org/10.1210/en.2005-0440). The current manuscript does provide some evidence of details in that signaling pathway, but these are relatively incremental additions to the knowledge base and limits the impact of the manuscript.
2) Figures 1 & 4: MHC immunofluorescence should be labeled in the figure for clarity (maybe next to the images).
3) Figure 2 -7. The standard these days is to include data points in bar graphs, as done in Figures 1 and 4a.
4) Figure 3 is missing from the manuscript.
5) Section 2.5, lines 199-201. The authors’ assertion that the PA-induced loss of Myh4 promoter activity was “significantly recovered” with the expression MyoD-T/A is questionable in my view. It appears that with that construct, promotor activity is reduced to near zero already (in the Con) such that the addition of PA has little room for further reduction.
6) Figure 7. Please label the axis (Myh4 expression).
7) Section 2.6, lines 227-228. The authors argue that Ro31 partially diminished the suppression of Myh4 expression by PA. However, the data and statistical analysis supporting that argument are unclear to me. The statistical mark indicates an interaction between the 4 groups in figure 7-A, but what is the specific interaction? Which groups are different from which? Are the Ro31(-)-PA and Ro31(+)-PA groups statistically different in the ANOVA?
8) Figure 8 legend should be expanded to explain each of the components in the figure.
9) Section 4.4: Was Gapdh validated as an appropriate housekeeper for these experiments (did its ddCT change significantly between treatment groups)?
10) Section 4.7: were the reporter assays validated in any way? If so, how?
Author Response
Please see the attached file named 'Response to the reviewer 1'

Reviewer 2 Report
The authors of the manuscript aimed to investigate the effect of palmitic acid (PA) treatment on the expression of different isoforms of myosin heavy chain (MyHC) in C2C12 cells during differentiation. Moreover, the authors sought to elucidate potential molecular mechanism(s) responsible for alterations in fast-type MyHC expresssion in response to PA treatment. The main finding of the study is that PA selectively suppresses the expression of fast-type MyHC by modulating MyoD activity via phosphorylation by protein kinase C (PKC). Although the study is relevant to the field and the manuscript is clearly written, there are some comments that need to be addressed in order to improve the manuscript's quality.
Major comments:
1. Figure 3, presumably showing the expression of specific MyHC proteins, is missing.
2. It is not clear whether the authors measured the content of isoforms of MyHC proteins separetly or altogether? In the "Original images of Western Blotting" file we can see that the authors did not differentiated between MyHC isoforms. In the Western blotting sub-section (4.5) the authors stated that they used "antibody against MHC (1:1000; MF-20; Hybridoma Bank, Iowa, IA, USA), which recognizes all MHC isoforms". At the same time, on page 4 (line 132) the authors wrote that "MHC I protein expression was also reduced by PA" and on page 9 (line 321) the authors wrote "protein expression of fast-twitch MHC". Please explain how did you measure the content of different MyHC isoforms by western blotting.
3. In the "Original images of Western Blotting" file there are 2 clear bands. Did you measure the optical density of both bands or not? What protein/isoform corresponds to each band?
4. Upon 4-day and 6-day differentiation (days of analysis), did the authors observe similar or distinct cellular morphology, differentiation markers etc. between control and PA-treated cells? The manuscript would benifit from showing changes (if any) in morphological parameters (diameter, area, width and length of C2C12 cells) and expression of differentiation/fusion markers in response to PA treatment.
Minor comment:
1. Page 3, line 100: "PA primarily inhibits the expression of MHC IIb in differentiated cells" will look better.
Round 2
Reviewer 1 Report
The reviewer appreciates the author's responses which have improved the manuscript.
Reviewer 2 Report
The authors satisfactorily addressed my comments.